# Awake Bruxism Identification: A Specialized Assessment Tool for Children and Adolescents—A Pilot Study

**DOI:** 10.3390/ijerph22070982

**Published:** 2025-06-22

**Authors:** Núbia Rafaela Ribeiro-Araújo, Anna Cecília Farias da Silva, Camila Rita Vicente Marceliano, Maria Beatriz Duarte Gavião

**Affiliations:** 1Departamento de Saúde Coletiva, Odontopediatria e Ortodontia, Faculdade de Odontologia de Piracicaba, Universidade Estadual de Campinas, Piracicaba 13414-903, SP, Brazil; n220881@dac.unicamp.br (N.R.R.-A.); a249194@dac.unicamp.br (A.C.F.d.S.); c229495@dac.unicamp.br (C.R.V.M.); 2Instituto de Ciências da Saúde, Faculdade de Odontologia, Universidade Federal do Pará, Belém 66075-110, PA, Brazil

**Keywords:** awake bruxism, bruxism identification, bruxism in children and adolescents, momentary ecological assessment

## Abstract

Awake Bruxism (AB) is defined as masticatory muscle activity during wakefulness, characterized by repetitive or sustained tooth contact and/or the bracing or thrusting of the mandible. AB remains less understood than Sleep Bruxism (SB), and its identification remains a methodological challenge. The aim of this study was to introduce the Awake Bruxism Identification Tool (ABIT), developed for children and adolescents aged 8 to 12 years, to facilitate the identification of AB. The tool integrates data from self-reports, clinical examinations, and the Ecological Momentary Assessment (EMA). It comprises questionnaires using a five-point Likert scale, an analog EMA component involving color-based responses, and a clinical inspection. The tool adopts the concept of an “AB Spectrum”, as it generates individualized scores based on the combined outcomes of these multiple assessment components. The ABIT was piloted with ten families to evaluate its comprehensibility, applicability, and reliability. The results demonstrated that the participants found the questions understandable, that the tool had a minimal impact on daily family routines, and that it required approximately 5–10 min to complete. Additionally, the test–retest reliability indicated temporal stability. In terms of identification, four children were classified within the “AB identified by report and self-report,” while three were identified through the “report, self-report, and EMA.” Based on participant feedback, adjustments were made to the instrument, including the addition of an item addressing Sleep Bruxism. Although the ABIT is being applied for the first time in a research setting, it presents a promising, clinically relevant approach grounded in the self-perception of children and their caregivers.

## 1. Introduction

Bruxism is understood as a motor behavior and can be classified into two distinct forms: Sleep Bruxism (SB) and Awake Bruxism (AB) [1]. SB is characterized as masticatory muscle activity during sleep that can be rhythmic (phasic) or non-rhythmic (tonic) and is neither a movement disorder nor a sleep disorder. AB, on the other hand, is masticatory muscle activity during wakefulness, characterized by repetitive or sustained tooth contact and/or the bracing (act of rigidly holding or stabilizing the mandible by applying sustained muscle contraction without tooth contact,) or thrusting (is the forward or lateral movement of the mandible involving active displacement of the jaw) of the mandible and is also not considered a movement disorder [2].

Bruxism is a general term that encompasses a broad spectrum of jaw muscle activities of various etiologies. It can be a sign of underlying disorders, a risk factor for clinical consequences, or possibly just a behavior without any pathological relevance [1,3]. It has a multifactorial etiology, involving biological and psychosocial factors [4,5]. Thus, it may be interpreted as a risk factor, a protective factor, or a neutral factor, depending on the specific clinical context [2].

Specifically regarding AB, its negative impact on health is evident, particularly in relation to Temporomandibular Disorders (TMDs) and their triggering or aggravating factors [6], as well as in relation to parafunctional habits and anxiety traits in children [7]. The sustained nature of muscle contraction in AB may explain the stronger association between AB and TMD [6], with increased muscle fatigue and pain resulting from bracing and thrusting [2]. It may also act as a risk factor for mechanical tooth wear [1].

These considerations highlight the ongoing progress in the scientific understanding of bruxism, which has culminated in the development of specific guidelines established by the so-called “Bruxism Consensus” [2,8,9]. In addition, standardized instruments have been developed for the assessment of bruxism, among which the Standardized Tool for the Assessment of Bruxism (STAB) [10,11] and the BruxScreen [12] stand out. These complementary tools represent important advances in the contemporary approach to bruxism, which is currently understood as an activity with a multifactorial etiology.

In this context, the Ecological Momentary Assessment (EMA) has been incorporated into studies on Awake Bruxism [3,13,14,15,16,17]. In these investigations, the use of the BruxApp^®^ (2.4.0 and 2.6.4 advanced version) application [3] enables data collection in natural environments [18] and has established itself as a central methodology for real-time behavioral assessments.

On the other hand, in pediatric populations, the application of the EMA is limited due to ethical considerations related to the restricted use of technology and screen exposure. Consequently, available studies on AB in this age group have primarily relied on questionnaires and reports, investigating its associations with various factors such as bullying [19], sleep, chronotype profiles [20], and associated factors [21,22,23].

Regarding the prevalence of bruxism, data show a considerable variability depending on the geographic region. Zieliński and colleagues, in a systematic review conducted in 2024 [24], reported a global prevalence of approximately 22.0% for overall bruxism (AB and SB) and 23.0% specifically for AB. Regional variations were also noted, with prevalence rates of 30.0% in South America, 25.0% in Asia, and 18.0% in Europe. Among children, the authors identified a prevalence of 9.0% for SB and a range between 6.0% and 11.0% for AB. Notably, a particularly high prevalence of 24.0% was reported among children in South America, accompanied by a wide confidence interval [24]. Specifically, studies on AB report diverse results, such as 37.3% [25], 4.1% [26], 20.1% [27], 38.4% [16], between 16.0% and 32.0% [28], 51.6% [22], and 51.1% [20].

The variation in prevalence estimates of Awake Bruxism indicate the need for standardized and effective instruments for its identification. In this context, the Assessment Bruxism Identification Tool (ABIT) was developed in 2023 for use in children and adolescents, based on the prevailing consensus at the time. Subsequently, the tool was adjusted according to the hierarchical model proposed in the most recent international consensus [2].

The development of the ABIT aims to provide an initial screening tool for the identification of AB, emphasizing family observation, self-perception, low costs, and simplicity to promote participant adherence. Accordingly, the present study aims to introduce the ABIT (preliminary version) and describe a pilot study conducted using its application, proposed as a feasible alternative in pediatric and adolescent clinical practice through a methodology based on self-reports, clinical examinations, and a playful, analog version of the Ecological Momentary Assessment (EMA), adapted for children and adolescents.

## 2. Materials and Methods

For the development of the ABIT, a comprehensive literature review was conducted on the concept, classification, and identification methods of AB [10,11,29,30,31,32] with particular emphasis on studies dedicated to the development of assessment tools.

ABIT is intended for children aged 8 to 12 years, with the minimum age set at 8 to ensure adequate comprehension of the questions. Its primary aim is to enhance the accuracy of AB recognition. In alignment with the current consensus on bruxism, which discourages the use of the term “diagnosis” [2], the tool is based on the concept of the “AB Spectrum,” interpreted from reports, self-reports, analog EMA, and clinical examination. This approach reflects individualized behavioral patterns over a given period. A key distinguishing feature of ABIT is its emphasis on self-perception, made possible through prior guidance provided to parents and caregivers [9].

The ABIT is recommended for use in clinical and academic settings, including pediatric dentistry offices, specialized clinics, and university-based clinics. However, its application in population-based epidemiological studies may be limited due to the need for individualized administration. In cases where the tool indicates a positive result, a complementary assessment using standardized protocols, such as the STAB, is recommended.

### 2.1. Tool Presentation

The ABIT was developed to be applied in two stages, initially in Brazilian Portuguese. The tool comprises the following components: Parents’ Report (R1), self-report of children and adolescents (SR), clinical assessment (CA), Parents’ Report after Guided Observation (R2), and Ecological Momentary Assessment (EMA). Each of these components will be detailed below.

Item 1 (Report 1-R1) is composed of three questions that must be answered by the guardian during the initial interview to assess the perception of the child’s/adolescent’s behavior in relation to AB. The first question assesses whether teeth grinding while awake has been observed: (1—Is the child’s jaw repeatedly moved sideways and/or back and forth while the teeth remain in contact?). The second question assesses whether teeth clenching has been noticed: (2—Are the upper teeth brought into contact with the lower teeth in a clenching motion?). The third question evaluates whether the tension in the facial muscles or if the jaw is held in a forward position has been observed: (3—Is it noticed whether the facial muscles appear tense or rigid (contracted) or if the chin is held in a forward (protruded) or sideways (lateral) position?). To facilitate this process, information is provided to guardians on how to identify AB, using illustrative images and real-life examples.

The response options for the three questions in Item R1 are presented as follows: “never,” “almost never,” “sometimes,” “most of the time,” “always,” and “don’t know,” with respective scores of 0, 1, 2, 3, and 4 assigned. The R1 score in the ABIT is calculated by summing the scores assigned to the three questions. The “don’t know” option is recorded as “DK” and is not assigned a score. If only one behavior among the three possible is reported by the participant, and the reported frequency is “almost never” (score 1 on the Likert scale) [33], the result is considered null, and the report is not classified as positive.

Item 2 (Self-Report—SR) is composed of three questions that are directed at the participating child/adolescent and are intended to assess their self-perception of AB. These questions must be answered during the initial interview. In the first question, the participants are asked whether they experience teeth grinding while awake: (1—Is your jaw repeatedly moved from side to side and/or back and forth while your teeth are kept in contact?). In the second question, the participants are asked whether teeth clenching is noticed: (2—Is it noticed that your upper and lower teeth are clenched together?). The third question asks whether facial muscle contraction or jaw protrusion has occurred: (3—Is it noticed whether their facial muscles become tense or rigid (contracted), or if the chin is positioned forward (protruded) or to the side (lateral)?

The response options for the three questions in Item SR are the same as those in Item R1, as is the scoring method for SR in the ABIT. Likewise, the interpretations for “DK” and null results follow the same criteria.

Item 3 (Clinical Assessment—CA) comprises information obtained from both the extraoral (ECA) and intraoral (ICA) clinical assessments, which involve the examination of the masticatory muscles, soft tissues, and teeth. This item must be completed by the researcher/dentist.

Each clinical feature identified in the ICA and ECA is marked as “Yes,” and each “No” is assigned 1 point in the CA score. In the ABIT protocol, a CA is considered positive only when both ECA and ICA present at least one positive finding for the same child or adolescent. However, the scores from both assessments are summed to define the spectrum of AB.

During the ECA, the masseter and temporal muscles (right and left) are examined through visual inspection and palpation. The presence of hypertrophy is recorded as Yes (Y) or No (N). In the ECA, the same scoring criteria are applied, with 1 point assigned for the presence of masseter or temporalis muscle hypertrophy. The interpretation of masseter hypertrophy is determined based on three conditions that justify a “Yes” response: hypertrophy that is both palpable and visible during forced occlusion, visible hypertrophy with a prominent mandibular angle, and visible hypertrophy accompanied by exostosis at the mandibular angle. For the temporal muscle, the presence of hypertrophy is assessed based on whether it is palpable and visible both during forced occlusion and at rest.In the ICA, the jugal mucosa, tongue, and buccal mucosa are examined. All findings are to be recorded as Yes (Y) or No (N). In the ICA, 1 point is assigned for the presence of buccal hyperkeratosis (linea alba), labial hyperkeratosis, labial indentation, tongue indentation, and buccal indentation. Tooth wear is assessed and noted as complementary data and does not participate in the diagnosis. For this record, a “five-point ordinal scale” is used: Grade 0 = no visible wear; Grade 1 = visible enamel wear; Grade 2 = visible wear with dentin exposure and loss of clinical crown height ≤ 1/3; Grade 3 = loss of clinical crown height >1/3 but <2/3; and Grade 4 = loss of clinical crown height ≥ 2/3. Care is taken to distinguish wear from chemical erosion, following specific criteria [34,35].

Item 4 (Report 2—R2) is composed of three questions, identical to those included in Item 1, but phrased using the past-tense verbs. These questions are addressed to the guardian and are to be answered following a period of observation of the child/adolescent, which is conducted over a surveillance interval of seven days, after appropriate guidance has been provided.

The response options for the three questions in Item SR are the same as those in Items R1 and SR, as is the scoring method for SR in the ABIT. Likewise, the interpretations for “DK” and null results follow the same criteria.

Item 5 (EMA) is related to the child’s/adolescent’s self-perception of behavior, and it is to be documented within seven days following the initial interview. This analog method is conducted using pencil/pen and paper, in which illustrations (Figure 1) of “emoji” vectors (representing emotional expressions) must be colored in by the child if any AB events are identified. To support this process, guidance is provided to the children on how to recognize AB, using illustrative images and real-life examples. Instructions for the at-home phase are as follows: the drawing sheet is to be kept with the child for seven days and must be affixed to a wall or piece of furniture using adhesive tape (to aid memory and prevent loss of the sheet) in the room where the child spends most of their time (typically the living room or bedroom). During school hours, the sheet is to be kept in the child’s notebook or backpack. Parents are advised to inform teachers and request permission for the child to carry the sheet during class. Caregivers are instructed to retrieve the sheet after the seven-day period. The sheet is composed of forty-two vector illustrations of emoji faces—six illustrations for each day of the week. One illustration is to be colored for each identified occurrence of AB. The cut-off point in ABIT for EMA to be considered positive for AB is four days or more with paintings present (EMA > or = 4)

### 2.2. Cut-Off Points and Data Interpretation

For data analysis, the aggregated item scores allow for the classification of the child or adolescent within the “AB Spectrum”, according to the following categories: “Not AB,” “AB based on report/self-report,” “AB based on report, self-report, and EMA”, “AB based on EMA,” and “AB based on clinical assessment”. When stronger evidence of the behavior and potential clinical consequences is observed, categories may be combined to reflect a more pronounced spectrum.

The “Not AB” classification corresponds to zero or null scores. R1, R2, and SR receive a zero score when the answers are “Never,” and a null score if only a single item is marked as “Almost Never” (score 1). The EMA record receives a zero score if no painting is recorded and a null score if the painting is below the cut-off point (less than four days with painting records). Null values are indicated in the tables as “Null” (R1/R2/AR) and “<4” (EMA).

“AB based on report/self-report (AB based on R/SR)” refers to scores higher than zero on the Likert scale [33] of R1, R2, and SR. “AB based on report, self-report, and EMA (AB based on R/SR and EMA)” corresponds to the correlation between “AB based on R/SR” and a positive EMA.

“AB based on clinical assessment (AB based on CA)” is characterized by the combination of positive ICA and ECA scores. A score higher than 0 on the ICA is recorded in the presence of indentations and/or hyperkeratosis (on the lips, buccal mucosa, and tongue), with each occurrence corresponding to an increase of one score point. A score greater than 0 on the ECA is recorded in the presence of masseter or temporal hypertrophy, with each occurrence resulting in an increase of 1 point. Tooth wear is recorded as complementary data and does not serve an identification purpose (the ordinal tooth wear scale) [19,22].

Thus, ABIT defines parameters based on cut-off points: each item has a score, clinical interpretation, and maximum spectrum. The sum of the spectra results in the Awake Bruxism Spectrum (AB Spectrum), which ranges from 0 to 96 (Table 1).

### 2.3. Pilot Study

This study represented the initial phase (Figure 2) of a broader investigation, which was submitted to and approved by the Research Ethics Committee of FOP-UNICAMP, Piracicaba, SP, Brazil (protocol code: 67100922.1.0000.5418, dated 18 May 2023). This study was conducted in accordance with the guidelines of the Declaration of Helsinki and informed consent was obtained from all subjects involved in the study.

The ABIT (Appendix A) was used for the first time with a group of 10 families who attended the Dental Specialties Center (CEO II, Piracicaba, SP, Brazil) of the municipal public health institution between June and August 2023. This group was selected based on the pediatric dentistry waiting list for that period, which included thirty-one children. Of these, sixteen were invited through invitations addressed to parents/guardians; ten accepted, three later dropped out, and five new invitations were extended until a convenience sample of ten participants was reached.

The inclusion criteria for the children were as follows: eight to twelve years old of either sex. The exclusion criteria included users of centrally acting drugs and/or patients with conditions that impaired their ability to understand and answer the questions. The inclusion criteria for parents were being the responsible companion during the initial interview and return visit. The exclusion criteria included illiteracy and/or conditions that impaired their ability to understand and answer the questions.

After acceptance, an anamnesis form was filled out (including the question about BS), and then the ABIT was applied by a single examiner (N.R.R.A) who had been properly trained (M.B.D.G.). Following this, a qualitative assessment of the ABIT items was conducted, focusing on comprehensibility, applicability, and temporal stability (Appendix A), using a questionnaire with three-point Likert Scale [33] response options.

Comprehensibility was assessed based on responses regarding how well the questions were understood (Table 1/Appendix A), and applicability (feasibility) was evaluated through questions about routine, ease/difficulty of participation, and the time taken to complete the form (Appendix A). Temporal stability (reliability) was assessed using a “Test–Retest” method in a group of five participants, with a retest conducted after fifteen days. A reliability analysis was conducted to assess the internal consistency of the questionnaire for parents. The agreement between the test–retest for ABIT scores was assessed using the Concordance Correlation Coefficient (CCC).

Participants were also asked for suggestions, and adjustments were made based on their feedback. The results of the qualitative ABIT test, AB frequency, and the comparison of test and retest data will be presented descriptively.

## 3. Results

### 3.1. Testing ABIT Items

In the evaluation of comprehensibility, six guardians said they fully understood the questions in the initial interview, while four said they partially understood. The recurring doubt referred to “whether the answers should take SB into account”. Regarding the household questions, all guardians reported fully understanding them.

The ten children selected the item “I understood everything” in relation to both the interview questions and the painting sheet task for home/school. One child expressed doubts about whether they could color the drawing later if the sheet was not available at the time. Despite this uncertainty, the child reported not having colored the drawing.

When assessing applicability, six guardians reported a “low level of difficulty” in their routines, while four stated there was “no difficulty at all”. All the children reported that it was easy to participate in the survey, both at home and at school. Five children selected the item “none” regarding the difficulty of participating at home, while the other five chose the item “almost none”. Regarding the difficulty of participating at school, seven children selected “none,” and three chose “almost none.” In terms of the time taken to complete the survey, all participants, both adults and children, selected the option “five to ten minutes.” As for suggestions for improvement, two mothers recommended that the home sheets be printed on a single page. The data cited are presented below (Figure 3).

The internal consistency of the instrument improved from the initial test (R1, Cronbach’s alpha = 0.462) to the retest (R2, Cronbach’s alpha = 0.827). Intraclass correlation coefficients indicated a high reliability between the two administrations, with values of 0.880 for consistency and 0.831 for agreement, demonstrating the good temporal stabilitof the instrument (Table 2).

### 3.2. AB Frequency in the Test Group

The results on the absolute frequency of AB refer to the individualization of the AB Spectrum. The complete data from the test (T) and retest (R) are presented in tables in the Figure 4 and Section A.1 and Section A.2.

The agreement between the test–retest for ABIT scores showed an almost perfect agreement (CCC = 0.997; 95% CI: 0.979–0.999), supporting the respective consistency.

The perceptions recorded (Table 3) reflect the type of AB event perceived (grinding, clenching, or bracing/thrusting) in both the Parents’ Reports and the children’s self-reports during the test and retest.

The findings related to dental wear (Appendix C) revealed that Grade 1 wear was identified in sixteen dental elements, Grade 2 in six elements, while Grades 3 and 4 were not detected in any teeth.

## 4. Discussion

### 4.1. About the Tool Development

The development of assessment instruments aims to support prevalence studies and to deepen the understanding of conditions that remain insufficiently clarified such as Awake Bruxism (AB), which involves multiple complex and not yet fully understood factors. This need becomes even more evident when considering the significantly larger number of studies focused on Sleep Bruxism (SB) [36]. The ABIT was designed to identify AB using a spectrum-based approach, as opposed to the traditional dichotomous classification of “present versus absent.” The instrument is targeted at children aged 8 to 12, an age group considered by the World Health Organization [WHO] [37] to be in transition between childhood and adolescence, but legally classified as childhood in Brazil, according to the Statute of the Child and Adolescent [38]. As a next step, it is recommended that a version be developed for adolescents aged 13 to 19, encompassing both middle adolescence (14–16 years) and late adolescence (17–19 years), in accordance with WHO classifications, to broaden the applicability of the instrument.

Instrumental methods, such as Electromyography (EMG), are considered the gold standard for identifying Awake Bruxism (AB). However, due to limitations in assessment dynamics and associated costs, a viable alternative is the incorporation of the EMA [18], which may be comparable to “device-based AB” [16]. Additionally, the use of parental reports, where caregivers observe and record daily behavior, is intended to reduce the reporting bias [5,21].

Accordingly, the ABIT was carefully designed to integrate reports, self-reports, and clinical evaluations into a scoring system for non-self-explanatory questions. This means the tool requires the researcher’s or clinician’s active involvement in training the respondent to recognize AB behaviors, as participants’ lack of awareness may affect the accuracy of the data [39], and parental education in recognizing AB represents a significant variable [40].

Nevertheless, during the initial phases of this project, there was concern regarding respondents’ understanding of the terms bracing and thrusting. However, after the careful formulation of the question in Brazilian Portuguese, it was determined that a pilot study could help clarify the context and guide potential adjustments. It was also acknowledged that the existing research on pediatric bruxism often relied on dichotomous questions (presence/absence), and that developing a tool that disaggregates AB-related behaviors into three separate questions could enhance the specificity of the behavioral identification.

Furthermore, to improve the comprehension among participants, especially children and adolescents, the term “chin” was intentionally used as a pedagogical strategy. Since the technical terms bracing and thrusting lack commonly understood equivalents in Brazilian Portuguese, their use might compromise the communication clarity. This adaptation aimed to facilitate understanding and ensure more accurate responses.

Additionally, the ABIT demonstrated an excellent overall internal consistency, with a Cronbach’s α value above 0.9. Specifically, the consistency for Report 1 (R1) was moderate (Cronbach’s α = 0.462), while for Report 2 (R2) it improved to a good level (Cronbach’s α = 0.827). This progression suggests that the observation period helped participants better understand the item, leading to more consistent responses.

The EMA, in turn, can improve the measurement accuracy [41], real-time self-perception [42], and awareness, in contrast to one-time self-reports [29,30]. This is particularly relevant given that AB is a behavior subject to fluctuations, and its identification through retrospective self-reports alone may fail to capture these variations [43]. Moreover, the frequency of AB measured via the EMA has been shown to remain stable over several months [44].

App-based EMA tools, such as BruxApp [45], generate detailed reports that allow the tracking of both the frequency and type of AB events, capabilities that are not available in the ABIT EMA. As an analog method, the ABIT’s EMA records participant’s perceptions but does not provide this level of detail.

However, in addition to generating reports, it was desirable to establish a cut-off point for the EMA, which was defined as “four affirmative responses” in a study that monitored the EMA concurrently with EMG [46]. Based on this, the ABIT EMA adopts the ≥ 4 threshold, as it is the only available alternative reference cut-off, even though it reflects data obtained through the virtual EMA. The analog-ludic EMA has proven to be feasible and appropriate [47] when conducted using paper and a pencil/pen [48], as it avoids screen exposure in children [49].

The behavior recording over seven consecutive days is consistent with response rates that showed no significant differences between weekdays and weekends [14]. Although it lacks comparative validation, a study combining an analog EMA with reminders/stickers and a digital EMA has already been documented [50].

Another key component of the ABIT is the clinical assessment (CA), which is based on both intraoral [19,51] and extraoral clinical markers [19,22,52]. AB is considered to be present based on the CA when at least one ICA and one extraoral marker ECA are observed simultaneously (ICA ≥ 1 and ECA ≥ 1). Since isolated markers lack sufficient predictive power, the identification of AB through the CA requires the concurrent observation of multiple clinical signs, which enhances the diagnostic accuracy.

In general, the clinical assessment focuses on the detection of muscle hypertrophy, myalgia [15], and soft tissue lesions, using a visual inspection and palpation. Although some studies rely on imaging techniques and photographic records, the ABIT offers a cost-effective alternative by prioritizing low-cost clinical methods.

Additionally, dental wear is considered a complementary finding [22], as it may result from physiological processes or past behaviors [53]. Therefore, it is not regarded as a definitive or exclusive marker for identifying AB.

The ordinal tooth wear scale was chosen for its simplicity, ease of application, and prior use in studies involving adolescents, such as those by [19,22]. Although the Smith and Knight Tooth Wear Index (TWI) represents a valid alternative, its clinical application and the criteria described in its glossary do not differ substantially from those of the adopted ordinal scale, which reinforces the methodological appropriateness of this choice. Other scales, such as the Basic Erosive Wear Examination (BEWE), are specifically designed for the assessment of dental erosion and are employed in studies focusing on this aspect [54] but have different scopes.

The ABIT aims to identify AB using simple clinical resources [12,41] following the recommendations of the STAB—Axis A [10,11], although it does not incorporate EMG due to costs and software limitations [5,14].

The studies by Manfredini and partners [10,11] and Lobbezoo and partners [12] significantly influenced the development of the ABIT, which was created between 2022 and 2023 based on the 2018 consensus, prior to the consolidation of the multidimensional STAB model. The ABIT aims to identify Awake Bruxism (AB) through an initial screening based on family self-perception, leveraging the advantages of the EMA [55,56,57]. It serves as a clinical screening tool. For cases classified within the “AB Spectrum,” a further evaluation is recommended using multidimensional protocols, such as the STAB and BruxScreen criteria, to enable a more comprehensive investigation [58].

The identification of AB in children and adolescents is a complex task [58]. The proposed scoring system serves as a strategy for the clinical interpretation of items [59], based on Classical Test Theory (CTT), which posits that “the sum of the items provides information about the individual” [60], although no studies have yet compared these data. Nevertheless, initial impressions of the tool suggest that it effectively reflects the construct (AB) it is intended to measure.

### 4.2. The Pilot Test of the Tool

The pilot experiment aimed to evaluate the ABIT items through the participant feedback and clinical interpretability analysis [61], employing the “probing” method [62]. The tool was well received, likely due to the research team’s approach in engaging with both parents and children, which is an essential factor for methodological reproducibility, given that parental reports are the most common data collection method in this type of study [10,40]. One of the main challenges was to maintain objectivity during the data collection; to address this, questions were formulated based on verifiable data, deliberately avoiding personal opinions or assumptions.

The comprehensibility assessment considered the vocabulary, sentence complexity, linguistic clarity, and contextual understanding of the questions. The ease of participant understanding was also evaluated, with results indicating no difficulties in responses and no missing data [63]. In terms of applicability, the ABIT proved to be a feasible tool for use in daily family routines, requiring only 5 to 10 min to complete; these findings align with the results of the BruxScreen study [12].

Moreover, the tool demonstrated reliability, as evidenced by consistent outcomes. Despite some variation in responses, the positioning on the “AB Spectrum” remained stable after retesting. On the other hand, adherence to the tool requires seven consecutive days of wake-time bruxism (AB) monitoring, as well as a follow-up visit to return the completed Report 2 (R2) and EMA records.

This requirement may represent a methodological limitation, as it depends heavily on participant motivation. Nonetheless, the clinical benefits for families seeking to identify bruxism in their children may outweigh the potential loss of participants due to dropout, offering at least a positive clinical return. Strategies such as family observation, the encouragement of self-awareness, low costs, and the simplicity of the tool further support user adherence. Additionally, qualitative data gathered through the applicability study indicated positive user feedback, particularly regarding the acceptance and feasibility of using the ABIT in the home setting.

Adjustments made to the tool included the addition of a single question on SB within the R1 and R2 items, aiming to reduce bias given that SB is more widely recognized [24]. Simultaneously, secondary data were collected, considering the possible positive association between AB and poorer sleep quality indicators [53] as well as evidence suggesting that SB and AB may predict one another [64,65].

The ABIT demonstrated an excellent overall reliability (Cronbach’s α > 0.9) and good temporal stability, with intraclass correlation coefficients (ICCs) of 0.880 for consistency and 0.831 for agreement. A significant improvement in the internal consistency was observed between the first (R1: α = 0.462) and second (R2: α = 0.827) reports, indicating that the observation period enhanced the participant understanding of the items. In addition, the agreement between the test–retest for ABIT scores showed an almost perfect agreement (CCC = 0.997; 95% CI: 0.979–0.999), supporting the respective consistency. These results suggest that the ABIT is a reliable and stable tool for AB screening, standing out as a clinically viable, low-cost instrument with potential for applications in family settings.

Based on this discussion, it is concluded that the ABIT was developed, tested, and appears to have fulfilled its objective as a simplified tool for identifying AB [31,66] who highlight the need for new instruments featuring effective, clinically oriented research pathways. This facilitates the implementation of knowledge on clinically relevant metrics.

### 4.3. The Frequency of AB in the Pilot Group

The frequencies observed in this pilot study, with 30.0% of children presenting “AB based on R/SR and EMA” and 40.0% with “AB based on R/SR,” demonstrate the initial feasibility of applying the ABIT. However, these data cannot yet be considered representative or directly comparable to findings from larger-scale studies. Additionally, it is important to highlight that the prevalence of AB in children may vary across different geographic regions. While global prevalence estimates for pediatric AB range from 6% to 11%, the rate observed in South America reaches 24.0% [24]. These findings suggest that regional factors, such as cultural and environmental influences, may contribute to such differences. Moreover, the methods of data collection and the tools employed can play a significant role in the discrepancies observed.

The implementation of multicenter studies using standardized methodologies is recommended to provide a broader and more accurate understanding of the prevalence and associated factors of AB in children.

Furthermore, studies indicate even higher AB rates among adolescents: 51.1% [20] and 51.6% [22] both focusing on youth aged 12 to 19 years. Among children aged 8 to 10, while in a systematic review with a meta-analysis [27], there was a reported population mean of 20.1% for AB in children. These data underscore the relevance of standardized and validated instruments to advance the epidemiological understanding of AB across different age groups.

To date, there are no reports of studies applying the EMA methodology specifically in children within the age group addressed in this work. Therefore, all cited studies refer to data collection via questionnaires relying on reports and self-reports as their methodology.

It is important to highlight that the AB identification in the present study was predominantly based on caregiver reports and children’s self-reports, thus it is classified as subject-based AB [2]. Although the EMA represents a more structured and real-time data collection method, it still depends on the individual’s response at the time of the application and is, therefore, a form of self-report. On the other hand, some studies classify the virtual version of the EMA as a type of device [16].

Thus, the term “identified by EMA” emphasizes the effort to increase precision in AB identification (adding a component to the composite scores and characterization of the AB Spectrum), given that isolated self-reports may be subject to questionable accuracy [46] and might offer insufficient certainty regarding AB frequency [67].

Positive EMA findings were consistent with positive parental reports, suggesting potential for the future validation of the analog EMA. Conversely, one child exhibited a positive self-report but null EMA results. Null EMA scores were also common in five children, indicating the perception of some “AB traits” on one or more occasions but below the cut-off point (<4) [46]. This suggests the possibility of sporadic, low-intensity AB, which could be more accurately analyzed through continuous monitoring without a reliance on specific cut-off points [61].

Regarding clinical markers (particularly hyperkeratosis, indentations, presence of linea alba, and masseter hypertrophy), which did not influence the AB Spectrum in this study, they should not be excluded from future analyses. Other studies [19,68] have observed clinical markers in children aged 8 to 11, including dental wear. Additionally, the data represent results from an initial test with a small convenience sample, making it essential to maintain the clinical assessment in an expanded study to evaluate ABIT’s reproducibility.

Another relevant aspect concerns the nature of the AB events identified. Teeth clenching was the most perceived event, followed by bracing or thrusting, and lastly teeth grinding. This pattern contrasts with [13], who found teeth grinding to be the most frequent event, followed by bracing, in a young population. Such differences may reflect variations in age groups, methodologies, or the participant awareness of oral behaviors. Although direct comparison is limited, these findings support current approaches investigating AB traits in natural environments [13].

Additionally, although there was no change in the AB Spectrum at the retest (Figure 4 and Table A1), there was a reduction in parental reports, self-reports, and EMA rates. It is difficult to determine which assessment moment most accurately reflects reality, though retest results may be more reliable due to a longer training period for observation. This may relate to increased awareness and behavioral re-education, as proposed by the Ecological Momentary Intervention (EMI) model [69]. Other hypotheses include reduced confounding between AB and SB, which may have influenced initial SB responses (Table A1), as well as participants possibly basing initial responses on past behaviors.

This study has some limitations: the potential occurrence of AB behaviors during times when the tool was inaccessible (e.g., bathroom, outings, parties, physical activities, classes, or exams); the complete dependence on participant cooperation and engagement, introducing a potential bias from adherence variability; and possible forgetfulness or distraction during the spontaneous analog EMA observation and recording.

Given the challenges in establishing non-instrumental methods for AB detection [51], the research team is considering developing a strategy such as a child-friendly watch or bracelet with a programmed random alarm or an auditory device worn as a bracelet. However, this proposal requires further refinement and evaluation after testing the tool in larger, more representative samples.

## 5. Conclusions

The present study described the ABIT, an innovative tool that, when tested in this pilot study, demonstrated a satisfactory comprehensibility, applicability, and reliability. By integrating scores from multiple sources, it identified children in the categories of subject-based AB, with additional support from EMA-based AB, under the adopted “AB Spectrum” concept, allowing for a sensitive approach to individual and temporal variations in behavior. Its main distinguishing feature lies in its focus on self-perception, encouraging the active participation of children and caregivers, provided they are properly guided and trained to recognize the behaviors being assessed. The ABIT is recommended for use in clinical and academic settings but is not suitable for epidemiological studies due to the need for individualized administration.

In summary, the ABIT presents itself as a tool for the more accurate identification of AB and, in combination with the playful analog EMA, demonstrates promising potential with reproducibility for expanded studies. Expectations for its translation and cross-cultural adaptation suggest it may significantly contribute to clinical research involving children and adolescents.

## Figures and Tables

**Figure 1 ijerph-22-00982-f001:**
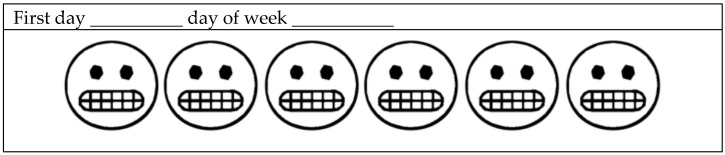
An example of playful drawings for one day of the week. The children must color the emoji when they notice AB on that day.

**Figure 2 ijerph-22-00982-f002:**
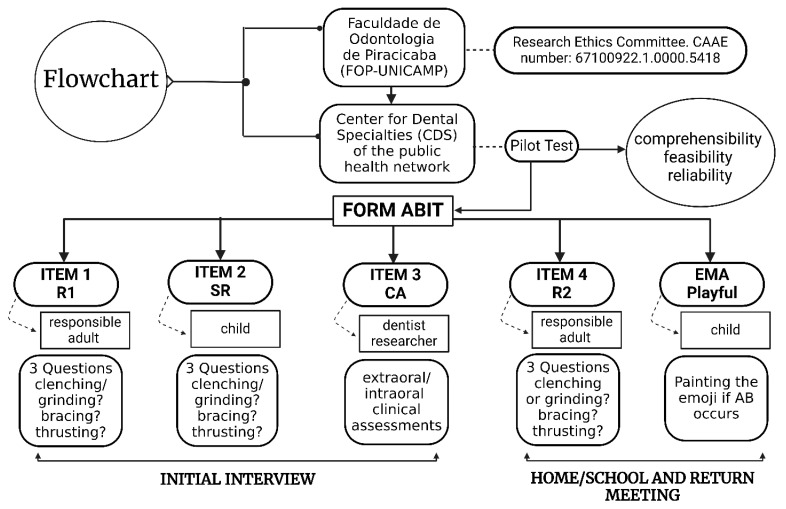
Study methodology flowchart.

**Figure 3 ijerph-22-00982-f003:**
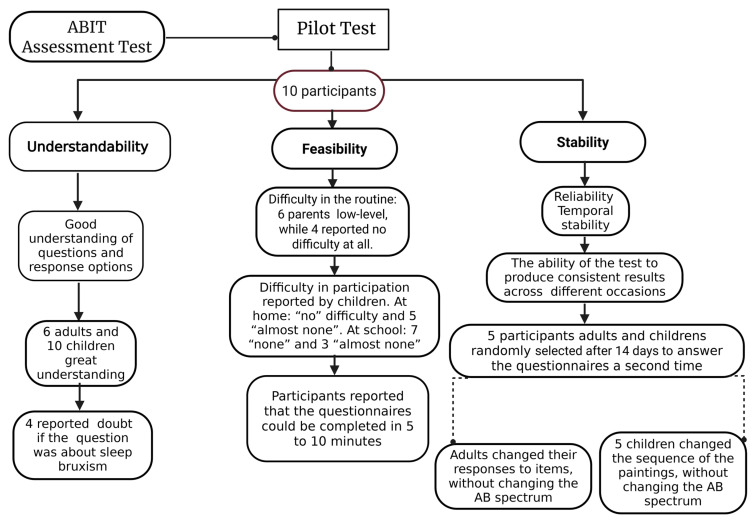
Pilot Test results on ABIT items.

**Figure 4 ijerph-22-00982-f004:**
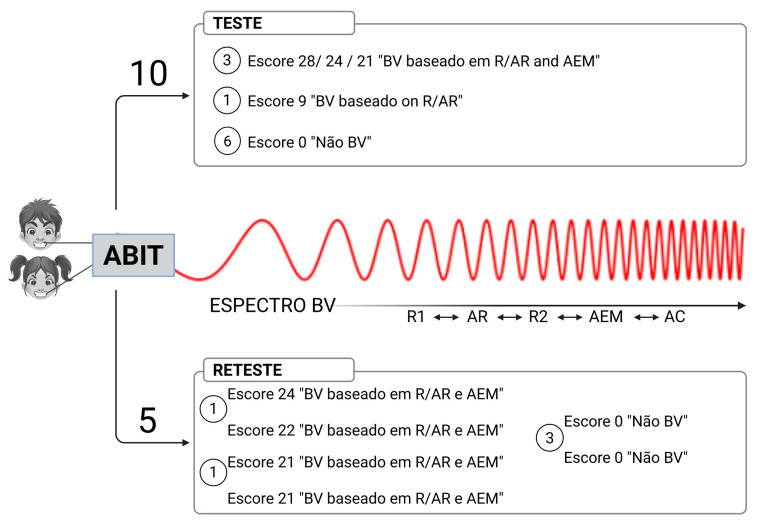
Spectrum AB results at test and retest.

**Table 1 ijerph-22-00982-t001:** Scoring and interpretation of ABIT items.

Item	Scores	Cut-off Values	Type of AB Identification	AB Spectrum
R1	0 a 4	Greater than or equal to 1 *	AB based on R/SR	12
SR	0 a 4	Greater than or equal to 1 *	AB based on R/SR	12
R2	0 a 4	Greater than or equal to 1 *	AB based on R/SR	12
ICA	0 a 6	Greater than or equal to 1 **	AB based on CA	6
ECA	0 a 2	Greater than or equal to 1 **	AB based on CA	2
EMA	0 a 42	Greater than or equal to 4 days with paintings	AB based on R/SR and EMA	42
				96

Note: EMA (Ecological Momentary Assessment), Score R1 (Parent’s Report 1), R2 (Parent’s Report 2), and SR (Child Self-Report); Likert Scale of 5 points. * value 1 is only considered valid if another Item scores higher than 0. ** value 1 is only valid if ECA (Extraoral Assessment Clinic) and ICA (Intraoral Assessment Clinic) are above 0.

**Table 2 ijerph-22-00982-t002:** Scale reliability statistics.

Cronbach’s α		Intraclass Correlation Coefficient	
		Consistency	Agreement
R1	0.462	0.880	0.831
R2	0.827		

Note: R1 (Report 1), R2 (Report 2).

**Table 3 ijerph-22-00982-t003:** Number of AB events that were reported in the test and retest.

Stage	AB Behavior	Reports R1 Parents	Reports R2 Parents	SR Children	Total
Test	Teeth Griding	1	0	4	5
Retest	Teeth Griding	0	0	3	3
Test	Teeth Clenching	1	5	5	11
Retest	Teeth Clenching	2	2	3	7
Test	Mandible Bracing/Thrusting	3	4	3	10
Retest	Mandible Bracing/Thrusting	2	2	1	5

Note: AB (Awake Bruxism), R1 (Parent’s Report 1), R2 (Parent’s Report 2), and SR (Child Self-Report). AB Behavior: List of behaviors that characterize AB: Griding|Clenching|Mandible Bracing/Thrusting.

## Data Availability

The datasets generated during and/or analyzed during the current study are available from the corresponding author upon reasonable request.

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
