# Peer review of "Awake Bruxism Identification: A Specialized Assessment Tool for Children and Adolescents—A Pilot Study"

_ijerph, 2025, doi:10.3390/ijerph22070982_

Round 1

Reviewer 1 Report

Comments and Suggestions for Authors

The article presents a tool for the assessment of a condition, awake bruxism, which really requires greater precision in diagnosis. The tool showed promising results, despite still being a pilot study. However, it is not clear whether the tool is an addition to existing diagnostic resources (with the aim of providing greater reliability in diagnosis) or whether it is intended to facilitate the diagnostic evaluation in order to reduce the steps. Is the instrument intended for use in epidemiological research or also for use in pediatric dentistry offices?

Author Response

Coments 1: The article presents a tool for the assessment of a condition, awake bruxism, which really requires greater precision in diagnosis. The tool showed promising results, despite still being a pilot study. However, it is not clear whether the tool is an addition to existing diagnostic resources (with the aim of providing greater reliability in diagnosis) or whether it is intended to facilitate the diagnostic evaluation in order to reduce the steps. Is the instrument intended for use in epidemiological research or also for use in pediatric dentistry offices?

Response 1: We sincerely appreciate your valuable feedback, which has significantly contributed to improving the quality of our manuscript. We would like to inform you that the following changes have been made in response to your suggestions:

A paragraph was added to the Methodology section detailing the characteristics of the ABIT tool and its recommended usage. This information was also reiterated in the Conclusion to reinforce the tool’s appropriateness for the intended context.

  • Lines 103-105:
    “Its primary aim is to enhance the accuracy of AB recognition. In alignment with the Current Consensus on Bruxism, which discourages the use of the term ‘diagnosis’…”
  • Lines 642-645:
    “ABIT is recommended for use in clinical and academic settings but is not suitable for epidemiological studies due to the need for individualized administration. In summary, ABIT presents itself as a tool for more accurate identification of AB…”

In response to your question about whether the tool is intended to complement existing diagnostic methods or to simplify the diagnostic process, we clarify the following:

“The tool presented in this study serves as a complement to traditional clinical identification methods, aiming to increase accuracy in recognizing bruxism. In accordance with the Current Consensus on Bruxism, which discourages the use of the term ‘diagnosis’ for this condition, we adopted the terminology ‘identification and recognition.’ It is a clinical tool specifically designed to identify the presence of bruxism, functioning as a screening or triage instrument. Its application does not involve a detailed or in-depth assessment of the condition. In cases where ABIT results suggest that the individual falls within the ‘AB Spectrum,’ complementary clinical evaluation is recommended, following multidimensional protocols such as STAB.”

Specification of the intended context of use for ABIT:
“The tool is intended for use in clinical research involving participant follow-up and is also suitable for pediatric and adolescent dental clinics. It is particularly useful in pediatric settings and in academic or training environments such as university clinics. ABIT was not designed for large-scale screening and, therefore, is not appropriate for epidemiological studies. This is due to its design for individualized evaluations, its complexity, and the need for specialized training, as well as the requirement for a home phase and follow-up visit, all of which make it unsuitable for large-scale studies.”

Reviewer 2 Report

Comments and Suggestions for Authors

Thank you for the opportunity to review this manuscript. I have the following comments:

To begin with, there is a significant issue with authorship. The MDPI submission system lacks one author who is listed in the manuscript, resulting in a discrepancy in the number of authors. This should be clarified with the editorial team.

It is also necessary to verify whether the co-authors' declarations are accurate and up to date.

Introduction:
Lines 32–35 – Citation no. 1: I would suggest referring more closely to citation no. 2, which reflects the updated consensus.

Line 54 – “with a recent global finding of 22.22%”: I recommend including the year of the study here, as it may not be considered "recent" in a few years. For example: “with a recent (2024) global finding of 22.22%”.

Line 54 – In addition, please provide epidemiological data for each continent based on citation no. 10. This is an important point that should be emphasised.

Line 56 – Throughout the introduction, I find the negative impact of bruxism is not sufficiently highlighted. This could be addressed in this paragraph or elsewhere, for example by noting that bruxism is associated with an increased risk of TMD. Please refer to the recent publication: “Global co-occurrence of bruxism and temporomandibular disorders: A meta-regression analysis”.

Furthermore, the introduction should include a description of the latest diagnostic tools for bruxism. The most recent ones are the BruxScreen (BruxScreener), followed by the BruxApp application. Please discuss these tools and explain how the authors’ research stands out in comparison.

Please also indicate in the title that this is a pilot study. Suggested title:
“Awake Bruxism Identification: A Specialised Assessment Tool for Children and Adolescents – Pilot Study”

I also noticed that repeatability analysis is included. Please clarify whether there are statistically significant differences between test administrations, and consider conducting appropriate statistical tests. Additionally, I recommend calculating the Intraclass Correlation Coefficient (ICC).

Finally, I suggest including, at the end of the manuscript, a calculation of the minimum sample size the authors plan to use in future studies. Please include power, effect size, and any other relevant parameters that the authors intend to base this on.

Kind regards,

Author Response

Comments 1: To begin with, there is a significant issue with authorship. The MDPI submission system lacks one author who is listed in the manuscript, resulting in a discrepancy in the number of authors. This should be clarified with the editorial team.

It is also necessary to verify whether the co-authors' declarations are accurate and up to date.

Response 1: 

  • The issue regarding a discrepancy in the number of authors due to the omission of Camila Rita Vicente Marceliano in the MDPI submission system has been resolved.
  • Camila Rita Vicente Marceliano has now been registered in the SciProfile system.

Clarification:

  • We confirm that all co-author declarations are accurate and up to date.

.

Comments 2: 

Lines 32–35 – Citation no. 1: I would suggest referring more closely to citation no. 2, which reflects the updated consensus.

Response 2: The definition of bruxism was updated in line with the latest consensus published on May 1, 2025 (Verhoeff et al., 2025), as follows:

SB is characterized as masticatory muscle activity during sleep that can be rhythmic (phasic) or non-rhythmic (tonic) and is neither a movement disorder nor a sleep disorder. AB, on the other hand, is masticatory muscle activity during wakefulness, characterized by repetitive or sustained tooth contact and/or bracing (act of rigidly holding or stabilizing the mandible by applying sustained muscle contraction without tooth contact,) or thrusting (is the forward or lateral movement of the mandible involving active displacement of the jaw) of the mandible, and is also not considered a movement disorder [2]. (Lines 38-35)

Comments 3: Line 54 – “with a recent global finding of 22.22%”: I recommend including the year of the study here, as it may not be considered "recent" in a few years. For example: “with a recent (2024) global finding of 22.22%”. Line 54 – In addition, please provide epidemiological data for each continent based on citation no. 10. This is an important point that should be emphasised.

Response 3:

The term “recent” was removed, and the year of the study was added for clarity: “with a global finding (2024) of 22.22%” (Zieliński et al., 2024). Additional epidemiological data were added by continent, based on Zieliński et al. (2024):

“…and 23.0% specifically for AB, with regional variations of 30.0% in South America, 25.0% in Asia, and 18.0% in Europe." (Lines 80-81)

Comments 4: Line 56 – Throughout the introduction, I find the negative impact of bruxism is not sufficiently highlighted. This could be addressed in this paragraph or elsewhere, for example by noting that bruxism is associated with an increased risk of TMD. Please refer to the recent publication: “Global co-occurrence of bruxism and temporomandibular disorders: A meta-regression analysis”.

Response 4: The negative health impacts of AB, particularly regarding its association with temporomandibular disorders (TMD), were added based on the article suggested (Zieliński et al., 2025).

It has a multifactorial etiology, involving biological and psychosocial factors [4] [5]. Thus, it may be interpreted as a risk factor, a protective factor, or a neutral factor, depending on the specific clinical context [2].

Specifically regarding AB, its negative impact on health is evident, particularly in relation to Temporomandibular Disorders (TMD) and their triggering or aggravating factors [6], as well as in relation to parafunctional habits and anxiety traits in children [7]. The sustained nature of muscle contraction in AB may explain the stronger association between AB and TMD [6], with increased muscle fatigue and pain resulting from bracing and thrusting [2]. It may also act as a risk factor for mechanical tooth wear [1].

Comments 5: Furthermore, the introduction should include a description of the latest diagnostic tools for bruxism. The most recent ones are the BruxScreen (BruxScreener), followed by the BruxApp application. Please discuss these tools and explain how the authors’ research stands out in comparison.

Ressponse 5: A discussion of the most recent diagnostic tools was added to the introduction and discussion. This includes STAB, BruxScreen, and BruxApp.

In addition, standardized instruments have been developed for the assessment of bruxism, among which the Standardized Tool for the Assessment of Bruxism (STAB) and the BruxScreen stand out. (Lines 60–63)

In these investigations, the use of the BruxApp® application [3] enables data collection in natural environments [18] and has established itself as a central methodology for real-time behavioral assessment. (Lines 67–69)

The use of BruxApp® for ecological momentary assessment (EMA) was discussed in comparison to ABIT.

App-based EMA tools, such as BruxApp [45], generate detailed reports that allow tracking of both the frequency and type of AB events, capabilities that are not available in the ABIT EMA. As an analog method, ABIT's EMA records participants' perceptions but does not provide this level of detail. (Lines 458-461).

The historical development of ABIT and its relationship to the STAB and BruxScreen models were clarified, emphasizing its original design based on the 2018 consensus.

The ABIT aims to identify AB using simple clinical resources [51] [41] following the recommendations of the STAB—Axis A [10] [55], although it does not incorporate EMG due to cost and software limitations [5] [56].

The studies by Manfredini et al. [55,57] and Lobbezoo et al. [51] significantly influenced the development of ABIT, which was created between 2022 and 2023 based on the 2018 consensus, prior to the consolidation of the multidimensional STAB model. ABIT aims to identify awake bruxism (AB) through an initial screening focused on family self-awareness, functioning as a clinical screening tool without detailed assessment. For cases classified within the “AB Spectrum,” complementary evaluation using multidimensional protocols such as STAB and BruxScreen criteria is recommended for a more thorough investigation. (Lines 494-504)

Comments 6: Please also indicate in the title that this is a pilot study. Suggested title:
“Awake Bruxism Identification: A Specialised Assessment Tool for Children and Adolescents – Pilot Study”

Response 6: The suggested revision to the title has been accepted with minor adjustments.

Comments 7: I also noticed that repeatability analysis is included. Please clarify whether there are statistically significant differences between test administrations, and consider conducting appropriate statistical tests. Additionally, I recommend calculating the Intraclass Correlation Coefficient (ICC).

Response 7: The reliability analysis was expanded, and statistical tests were applied to examine differences between test administrations.

The following metrics were included: Intraclass Correlation Coefficient (ICC), Cronbach’s α for R1 and R2 (caregiver reports), and test-retest consistency.

A reliability analysis was conducted to assess the internal consistency of the questionnaire. (Lines 284-285)

The internal consistency of the instrument improved from the initial test (R1, Cronbach's alpha = 0.462) to the retest (R2, Cronbach's alpha = 0.827). Intraclass correlation coefficients indicated high reliability between the two administrations, with values of 0.880 for consistency and 0.831 for agreement, demonstrating good temporal stability of the instrument. Lines 328-332)

Tables were included to summarize ICC and Cronbach's α statistics.

Comments 8: Finally, I suggest including, at the end of the manuscript, a calculation of the minimum sample size the authors plan to use in future studies. Please include power, effect size, and any other relevant parameters that the authors intend to base this on.

Response 8: At present, no finalized study design based on a target variable is available to support an exact sample size calculation. However, we anticipate using a statistical power above 0.80 in future studies. Effect size will be determined according to the statistical tests to be applied, such as group comparisons or correlation analyses.

Reviewer 3 Report

Comments and Suggestions for Authors

Dear Authors,
I read your manuscript titled "Awake Bruxism Identification: A Specialized Assessment Tool for Children and Adolescents."with great interest. 
It is clear that considerable thought and effort have gone into developing a novel and potentially impactful tool for the assessment of awake bruxism in children and adolescents. Your work addresses a critical gap in the field and introduces an innovative approach to diagnosis.
That said, the manuscript would benefit from editorial refinement to enhance its readability and focus. Below are several suggestions that may assist in improving the clarity and overall quality of the paper:

Title:
•    As this is a pilot study involving only 10 families, it may be appropriate to reflect this in the title by including the term "pilot study."

Introduction:
•    The section provides a very detailed overview of the bruxism consensus; however, it may be overly elaborate. Consider streamlining this part.
•    Please note that the consensus has recently been updated. The previous classification into "possible," "probable," and "definite" bruxism is no longer widely accepted. You may wish to refer to the newer terminology (e.g., patient-based, clinical-based).
•    There is limited discussion of bruxism in children, which is central to your study. Consider expanding on the prevalence, etiology, and the existing gap in diagnostic tools for pediatric populations. The STAB tool, which is discussed later, should be introduced earlier in the manuscript.

Materials and Methods:
•    Line 74: Please clarify the minimum age for self-reporting.
•    R1, Question 3 (Line 94): The question regarding parental observation of muscle tension may be problematic. Such observations are subjective and potentially unreliable compared to more visible behaviors like protrusive movements. Consider discussing the internal validation of this item.
•    Children and adolescents should not necessarily be assessed using the same approach. While the use of colored facial expressions is creative and engaging for younger children, it may be perceived as inappropriate or overly simplistic by adolescents. Older participants might prefer to self-report clenching or grinding behaviors in writing.
•    Tooth wear: Is there a different grading system for mixed dentition, particularly for children aged 8–12?

Results:
•    Line 340: Please include percentages to complement the raw data.
•    Tables 2 and 4 could be significantly shortened. It may not be necessary to present individual participant data in full. Instead, consider summarizing the findings and including one example for illustration.

Discussion:
•    The discussion does not address expected compliance with the tool (e.g., tracking bruxism for 7 days, attending at least two dental visits). It would be helpful to include commentary on feasibility and adherence.
•    Consider discussing whether different age groups might require tailored versions of the tool.

Supplementary meterials:
•   are currently in Brazilian Portuguese. Please provide English translations so that non-Portuguese-speaking reviewers and readers can fully understand the materials.

Author Response

Comments 1: I read your manuscript titled "Awake Bruxism Identification: A Specialized Assessment Tool for Children and Adolescents."with great interest. 
It is clear that considerable thought and effort have gone into developing a novel and potentially impactful tool for the assessment of awake bruxism in children and adolescents. Your work addresses a critical gap in the field and introduces an innovative approach to diagnosis.
That said, the manuscript would benefit from editorial refinement to enhance its readability and focus. Below are several suggestions that may assist in improving the clarity and overall quality of the paper:

Response 1: We sincerely thank Reviewer 3 for the thoughtful and detailed feedback. Below, we provide point-by-point responses to each comment, along with a summary of the changes made to the manuscript. All updates have been highlighted in pink in the revised version of the manuscript.

Comments 2: Title: As this is a pilot study involving only 10 families, it may be appropriate to reflect this in the title by including the term "pilot study."

Response 2: Accepted. The title was revised as follows:
“Awake Bruxism Identification: A Specialized Assessment Tool for Children and Adolescents – Pilot Study.”

Comments 3: The section provides a very detailed overview of the bruxism consensus; however, it may be overly elaborate. Consider streamlining this part.
Please note that the consensus has recently been updated. The previous classification into "possible," "probable," and "definite" bruxism is no longer widely accepted. You may wish to refer to the newer terminology (e.g., patient-based, clinical-based).
There is limited discussion of bruxism in children, which is central to your study. Consider expanding on the prevalence, etiology, and the existing gap in diagnostic tools for pediatric populations. The STAB tool, which is discussed later, should be introduced earlier in the manuscript.

Response 3: We have revised the text to increase objectivity while retaining the necessary background to support the study’s rationale. The section was streamlined in Lines 58- 60.

The manuscript was updated with the most recent terminology from the 2025 consensus (Verhoeff et al., 2025). However, we retained some legacy terms for contextual purposes, as the tool was developed prior to the official release of the consensus (submitted on April 2, 2025). Nevertheless, we clarified the alignment with the updated patient-centered approach. (Lines 227–241; Table 1).

We included a new paragraph in the introduction and expanded the discussion to emphasize the lack of EMA-based studies in pediatric populations. Recent studies from 2022 to 2024 were cited to demonstrate prevalence and associated factors.
(Lines 70–78; 572–580)

Comments 4: Materials and Methods:
•    Line 74: Please clarify the minimum age for self-reporting.
•    R1, Question 3 (Line 94): The question regarding parental observation of muscle tension may be problematic. Such observations are subjective and potentially unreliable compared to more visible behaviors like protrusive movements. Consider discussing the internal validation of this item.
•    Children and adolescents should not necessarily be assessed using the same approach. While the use of colored facial expressions is creative and engaging for younger children, it may be perceived as inappropriate or overly simplistic by adolescents. Older participants might prefer to self-report clenching or grinding behaviors in writing.
•    Tooth wear: Is there a different grading system for mixed dentition, particularly for children aged 8–12?

Response 4:

The minimum age for self-reporting has been added to the methodology. The ABIT was designed for children aged 8–12, with the lower limit defined to ensure understanding. (Lines 102-103)

The internal consistency of R1 and R2 was analyzed using Cronbach’s α and ICC. These results were discussed in both the Results and Discussion sections, highlighting the importance of structured observation and training.
We added a paragraph suggesting future adaptations for adolescents aged 13–19 (Lines 416-422).

We used the ordinal tooth wear scale, which has been applied in similar-age populations. Its rationale and comparison to other indices (e.g., Smith and Knight TWI) are included.
(Methodology Lines 236-238, Discussion Lines 483–492)

Comments 5: Results:
•    Line 340: Please include percentages to complement the raw data.
•    Tables 2 and 4 could be significantly shortened. It may not be necessary to present individual participant data in full. Instead, consider summarizing the findings and including one example for illustration.

Response 5: Percentages were added to Section 4.3 on AB frequency. (Lines 564-565)

We summarized the tables and moved full individual-level data to the appendix. A statistical summary and an illustrative figure were included.
(Figure 4; Appendix A1 & A2)

Comments 6: Discussion:
•    The discussion does not address expected compliance with the tool (e.g., tracking bruxism for 7 days, attending at least two dental visits). It would be helpful to include commentary on feasibility and adherence.
•    Consider discussing whether different age groups might require tailored versions of the tool.

Response 6: Accepted. A dedicated paragraph was added to the Discussion, addressing the 7-day monitoring requirement, expected adherence, and participant feedback (Lines 416-422)

We included discussion on this point, recommending future versions adapted for older adolescents. (Lines 416-422).

Comments 7: Supplementary meterials:
•   are currently in Brazilian Portuguese. Please provide English translations so that non-Portuguese-speaking reviewers and readers can fully understand the materials.

Response 7: The ABIT and Pilot Test Tool were translated into English and included as supplementary files (not cross-culturally adapted). Supplementary Materials section updated accordingly.

Round 2

Reviewer 2 Report

Comments and Suggestions for Authors

Thank you for resubmitting the manuscript for review. After re-evaluating the revised version, I have the following comments:

  1. Referencing format – In the preparation of the manuscript, instead of writing citations as ‘‘[3] [13] [14] [15] [16] [17]’’, it should follow the format ‘‘[3,13–17]’’. Please ensure that the citation format is corrected throughout the entire manuscript.
  2. Lines 75–76 – The sentence: “The prevalence of bruxism in children and adolescents shows wide variation, with estimates ranging from 5.0% to 50.0% [24].” – is not supported by the most up-to-date evidence. According to recent findings, the prevalence of sleep bruxism (SB) is approximately 9%, while awake bruxism (AB) ranges between 6% and 11%. Please revise this accordingly and refer to the study with DOI: 10.3390/jcm13144259.
  3. Effect size in Table 3 – Please add the effect size to Table 3. It is unclear why this information was not included by the authors. Kindly revise this, and also include in the description the thresholds/interpretation ranges for effect sizes. The following references may be helpful in this regard: 10.11613/BM.2021.010502 / 10.1001/jamaoto.2023.0159

Author Response

Comments 1:  Referencing format – In the preparation of the manuscript, instead of writing citations as ‘‘[3] [13] [14] [15] [16] [17]’’, it should follow the format ‘‘[3,13–17]’’. Please ensure that the citation format is corrected throughout the entire manuscript.

Response 1: Thank you for your valuable comment. We agree. We have revised the citation format throughout the manuscript according to the recommended style. All changes are highlighted in gray, as in the following example: [3,13–17].

Comments 2: Lines 75–76 – The sentence: “The prevalence of bruxism in children and adolescents shows wide variation, with estimates ranging from 5.0% to 50.0% [24].” – is not supported by the most up-to-date evidence. According to recent findings, the prevalence of sleep bruxism (SB) is approximately 9%, while awake bruxism (AB) ranges between 6% and 11%. Please revise this accordingly and refer to the study with DOI: 10.3390/jcm13144259.

Response 2: Thank you for this comment. We agree. The manuscript has been revised. The text has been updated on lines 79 to 82, as well as on lines 549 to 554, based on the most recent evidence on prevalence from the cited study. These changes are highlighted in green.

Commnets 3: Effect size in Table 3 – Please add the effect size to Table 3. It is unclear why this information was not included by the authors. Kindly revise this, and also include in the description the thresholds/interpretation ranges for effect sizes. The following references may be helpful in this regard: 10.11613/BM.2021.010502 / 10.1001/jamaoto.2023.0159.

Response 3: Thank you for ponting this out. Therfore, after reviewing the results, we considered that this table was not reliable, due to small sample size (N=10), which was distributed as follows: No AB (n=6); AB based on R/SR (n=1); AB based on R/SR and EMA (n=3). A group with only one observation has no variability; therefore, it is not possible to estimate its variance, which is essential for statistical comparison tests. In this context, we consider that the descriptive data about sample distribution could be enough for the respective understanding (Figure 1, Table 3, and Appendix A1). Thus, Table 3 in the first revision was deleted. 

Moreover, the agreement between test-retest for ABIT scores was checked using Concordance Correlation Coefficient, showing good result. Item 3.2 (lines 365 a 369) was rewritten as follows:

3.2 AB frequency in the test group

The results on the absolute frequency of AB refer to the individualization of the AB spectrum. The complete data from the Test (T) and Retest (R) are presented in tables in Appendices (Figure 4 / Tables in Appendix A1 and A2).

The agreement between the test-retest for ABIT scores showed almost perfect agreement (CCC = 0.997; 95% CI: 0.979–0.999), supporting the respective consistency”

The text (lines 532 to 534) was modified to include a discussion of the consistency data.

These changes are highlighted in green.

*Note: We would like to inform you that some highlights in yellow correspond to typographical corrections identified by the authors.

Once again, we are grateful for your constructive feedback.